# Age-Related sncRNAs in Human Hippocampal Tissue Samples: Focusing on Deregulated miRNAs

**DOI:** 10.3390/ijms252312872

**Published:** 2024-11-29

**Authors:** Ainhoa Alberro, Rocío Del Carmen Bravo-Miana, Saioa GS Iñiguez, Andrea Iribarren-López, Marta Arroyo-Izaga, Ander Matheu, Maider Muñoz-Culla, David Otaegui

**Affiliations:** 1Neuroimmunology Group, Neuroscience Area, Biogipuzkoa Health Research Institute, 20014 San Sebastián, Spain; ainhoa.alberrogaritano@bio-gipuzkoa.eus (A.A.); rociodelcarmen.bravomiana@bio-gipuzkoa.eus (R.D.C.B.-M.); saioa.garcia-serranoiniguez@bio-gipuzkoa.eus (S.G.S.I.); andrea.iribarrenlopez@bio-gipuzkoa.eus (A.I.-L.); 2Neurodegenerative Diseases Research Area of CIBER (CIBERNED), Carlos III Health Institute (ISCIII), 28029 Madrid, Spain; 3BIOMICs Research Group, Lascaray Research Center, University of the Basque Country (UPV/EHU), Bioaraba, 01006 Vitoria-Gasteiz, Spain; marta.arroyo@ehu.eus; 4Cellular Oncology Group, Oncology Area, Biogipuzkoa Health Research Institute, 20014 San Sebastián, Spain; ander.matheufernandez@bio-gipuzkoa.eus; 5IKERBASQUE, Basque Foundation for Science, 48009 Bilbao, Spain; 6Frailty and Healthy Ageing Research Area of CIBER (CIBERfes), Carlos III Health Institute (ISCIII), 28029 Madrid, Spain; 7Department of Basic Psychological Processes and Their Development, University of the Basque Country (UPV/EHU), 20018 San Sebastián, Spain

**Keywords:** small non-coding RNA, miRNA, brain, hippocampus, aging, longevity, transcriptome regulation, centenarians

## Abstract

Small non-coding RNAs (sncRNAs), particularly microRNAs (miRNAs), play an important role in transcriptome regulation by binding to mRNAs and post-transcriptionally inhibiting protein production. This regulation occurs in both physiological and pathological conditions, where the expression of many miRNAs is altered. Previous reports by our group and others have demonstrated that miRNA expression is also altered during aging. However, most studies have analyzed human peripheral blood samples or brain samples from animal models, leaving a gap in knowledge regarding miRNA expression in the human brain. In this work, we analyzed the expression of sncRNAs from coronal sections of human hippocampal samples, a tissue with a high vulnerability to deleterious conditions such as aging. Samples from young (n = 5, 27–49 years old), old (n = 8, 58–88 years old), and centenarian (n = 3, 97, 99, and 100 years old) individuals were included. Our results reveal that sncRNAs, particularly miRNAs, are differentially expressed (DE) in the human hippocampus with aging. Besides, miRNA-mediated regulatory networks revealed significant interactions with mRNAs deregulated in the same hippocampal samples. Surprisingly, 80% of DE mRNA in the centenarian vs. old comparison are regulated by hsa-miR-192-5p and hsa-miR-3135b. Additionally, validated hsa-miR-6826-5p, hsa-let-7b-3p, hsa-miR-7846, and hsa-miR-451a emerged as promising miRNAs that are deregulated with aging and should be further investigated.

## 1. Introduction

Aging is a complex and multifactorial process characterized by a progressive decline in physiological functions and an increased vulnerability to diseases, ultimately leading to death [1,2,3]. Understanding the molecular mechanisms that drive aging is critical for developing strategies to promote healthy aging and treat age-associated diseases. In this field, one of the emerging research areas is the study of transcriptome regulation [4]. Small non-coding RNAs (sncRNAs), particularly microRNAs (miRNAs), play significant roles in regulating gene expression and have been previously associated with aging [5,6,7,8,9].

miRNAs are typically 20–24 nucleotides in length and exert their regulatory functions by binding to complementary sequences on target RNAs, including messenger RNAs (mRNAs), introns, and 3′ untranslated regions (UTRs), resulting in mRNA degradation and, at the post-transcriptional level, inhibiting the protein production [10,11,12]. Since their discovery in the early 1990s, miRNAs have been found to be implicated in several biological processes, including development, differentiation, cell proliferation, stress responses, and apoptosis, and they have also been shown to be implicated in the aging and longevity processes [11,13]. miRNAs can regulate specific target genes involved in fundamental cellular processes, and their deregulation could be a consequence of age-related cellular stress and senescence, representing a possible protective response or perhaps a cause [14,15,16]. Understanding the complex interactions between miRNAs and their target genes will be essential to unlock their full potential in aging and longevity.

In this context, previous reports by our group and others found that miRNA expression is also altered by physiological aging [5,17,18]. However, most studies analyzed human peripheral blood samples [5,18,19] or other cell models [20], as well as brain samples from animal models [19,21,22]. This leaves a significant gap in our knowledge of miRNA expression in the human brain. In this work, we evaluated miRNA expression in the human hippocampus –a biologically complex region associated with learning and memory formation and storage– from young, old, and centenarian samples as a model of healthy aging.

## 2. Results

### 2.1. Differentially Expressed miRNAs in Human Hippocampal Tissue with Age

A microarray analysis was performed to study the sncRNAs differentially expressed in hippocampal tissue samples (n = 16, Table 1). The results obtained from the differential expression analysis (fold change (FC) > |2| and *p*-value < 0.05) identified 10 differentially expressed (DE) sncRNAs (9 DE miRNAs) in old vs. young samples (Figure 1A), 11 DE sncRNAs (10 DE miRNAs) in centenarian vs. old (Figure 1B), and 20 DE sncRNAs (18 DE miRNAs) in centenarian vs. young (Figure 1C). Detailed information, including transcript ID, accession number, FC, and *p*-value obtained for each DE feature, are shown in Appendix A. Specifically, we found that among the 41 DE sncRNAs, 24 sncRNAs were upregulated in old vs. young (3 sncRNAs), in centenarian vs. old (8 sncRNAs), and in centenarian vs. young individuals (13 sncRNAs). Conversely, 17 sncRNAs were downregulated, with 7, 3, and 7 showing a decreased expression, respectively, in each comparison. Remarkably, a greater number of miRNAs were altered in centenarians compared to young individuals than in old relative to young subjects. In addition, 90% of the DE sncRNAs were miRNAs, and only one small nucleolar RNA (snoRNA), snR38A, was downregulated in the old vs. young group. A Venn diagram of the DE sncRNAs across the three comparisons showed one common miRNA (hsa-miR-4314) between centenarian vs. old and centenarian vs. young comparisons and six miRNAs between old vs. young and centenarian vs. young (Figure 1D).

Additionally, a correlation analysis was conducted between sncRNA expression and age to identify further relevant features. sncRNAs that present a positive or negative Pearson correlation coefficient (r > |0.6| and a *p*-value < 0.01) are shown in Appendix A. This analysis revealed 12 mature miRNAs that gradually increase or decrease with age (Figure 1E). Among these, six mature miRNAs were identified as being significant in both the differential expression and correlation analyses (Figure 1F).

### 2.2. miRNA-Mediated Regulatory Networks Revealed Significant Interactions with mRNAs Analyzed in the Same Hippocampal Samples

Taking advantage of the available data, we built interaction networks to conduct a comprehensive analysis of the transcriptional mRNA regulation mediated by DE miRNAs in the context of the human hippocampus. Briefly, the putative mRNA targets of the DE miRNAs were obtained from the miRTarBase database. Since the DE mRNA data of the same brain donors had been previously published (see [17], Saez-Antoñanzas et al., 2024), only interactions between DE mRNA targets regulated by the DE miRNAs in the same sample were considered in the networks. Finally, interactions exhibiting opposite regulation (with an FC > |1.2|) were plotted (Appendix A), i.e., those in which if the microRNA was upregulated, its mRNA target was downregulated, or vice versa.

For the old vs. young comparison, this analysis identified 2646 in silico potential interactions in the miRTarBase database, which were reduced to 1095 experimentally validated potential interactions and, finally, to 17 mRNAs showing an opposite regulation (FC >|1.2|) with six miRNAs in the studied samples (Figure 2A and Appendix A). In the centenarian vs. old comparison, a total of 4664 in silico potential interactions were identified. Of these, 2151 were experimentally validated potential interactions, and we found 47 interactions with opposite regulation, involving 44 mRNAs and six miRNAs (Figure 2B and Appendix A). In the centenarian vs. young comparison, there were 6044 in silico interactions, with 2598 experimentally validated in the database, and we reported 39 interactions with opposite regulation (which involved 36 mRNAs and 12 miRNAs) in our samples (Figure 2C and Appendix A). On the other hand, the list of miRNAs identified through the correlation analysis was used for the same analysis, revealing 4706 in silico potential interactions, 31 experimentally validated potential interactions, and, finally, 18 miRNA–mRNA interactions with opposite regulation found in the studied samples (17 mRNAs and 10 miRNAs). Of note, DE miRNAs and DE mRNA target nodes were colored based on the FC (Figure 2A–C) or the positive/negative correlation trend with age (Figure 2D). Changes in miRNA expression were linked to alterations in target mRNA levels. Surprisingly, the highest number of significant DE mRNA–miRNA interactions were observed in the comparison between centenarians and old individuals. Furthermore, most interactions are concentrated in only two miRNAs (hsa-miR-192-5p and hsa-miR-3135b). Finally, the mentioned 17, 44, 36, and 17 age-related mRNAs regulated by miRNAs in each comparison were used for a Gene Ontology (GO) analysis. However, no relevant GO information could be obtained from the shortlists of targeted mRNA.

### 2.3. hsa-miR-6826-5p, hsa-let-7b-3p, hsa-miR-7846-3p, and hsa-miR-451a Significantly Correlated with Age

After the analysis, six candidates were selected from those identified through differential expression and correlation analysis and subsequently validated in an independent cohort of 118 hippocampal RNA samples. These exhibited an FC >|2.5| and/or a Pearson correlation coefficient r > |0.6| and are shown in Appendix A, along with the type(s) of analysis through which they were identified.

hsa-miR-6826-5p was significantly upregulated in old compared to young individuals, with a trend to a higher expression in centenarian vs. young individuals (Figure 3A). Similarly, hsa-let-7b-3p was significantly upregulated in old compared to young donors, with a trend towards even higher expression in centenarians (Figure 3C). Other miRNA candidates did not show significant differences under the studied conditions. On the other hand, the correlation analysis in the validation cohort showed that hsa-miR-6826-5p, hsa-let-7b-3p, and hsa-miR-7846 have a significant positive correlation with age (Figure 3A,C,F) while hsa-miR-451a displayed a significant negative correlation with age (Figure 3E). However, hsa-miR-4441 and hsa-miR-1825 did not show significant age-related correlations (Figure 3B,D).

## 3. Discussion

The brain is a highly structured organ that contains billions of neurons, glial cells, and a vast vascular network. The hippocampus is a brain region vital for learning, memory, and emotional regulation, and it has been associated with age-related cognitive decline and the onset of dementia, presenting a pivotal role in the study of aging, longevity, frailty, and neurodegenerative diseases [23,24]. Our findings revealed that sncRNAs, and particularly miRNAs, are deregulated in the aged human hippocampus. To understand the potential effect of this deregulation and take advantage of the previously conducted transcriptomal analysis of the same samples (see [17], Saez-Antoñanzas et al., 2024), we built networks of the predicted interactions between the DE sncRNAs reported here and the DE mRNAs reported in the previous study. Interestingly, many of these DE mRNAs were oppositely targeted by more than one DE miRNA, such as SLC26A2, NFIC, TYRO3, and SPTLC3. It is noteworthy that four distinct DE miRNAs were predicted to interact with NFIC in the networks: Two miRNAs in centenarian vs. young and two miRNAs in centenarian vs. old. This gene encodes a DNA-binding protein that functions as a transcription factor. Its deregulation has been linked to central nervous system (CNS) diseases [25], like Alzheimer’s disease [26], emphasizing the crucial role of the intricate network of transcriptome regulation. The network analysis also underlines the importance of hsa-miR-192-5p and hsa-miR-3135b as major regulators, being able to control 80% of DE mRNA identified between centenarians and old individuals. hsa-miR-192-5p was upregulated in centenarians in comparison to old individuals, and it has been related to neural function repair and the promotion of neural apoptosis in murine models [26]. Conversely, hsa-miR-3135b was downregulated in centenarians compared to old individuals, and it has been identified as a regulator of the REL/SOD2 pathway in a mouse model of ischemic cerebral infarction [27]. In addition, there are reports associating this microRNA with several other diseases, such as severe hypertension, heart failure, and diabetes [28,29,30,31]. However, neither hsa-miR-192-5p nor hsa-miR-3135b was previously related to aging or longevity. Considering that age-related changes driven by miRNAs are largely tissue-specific [32], the precise biological functions of these miRNAs within the brain, particularly in the hippocampus, should be further investigated.

On the other hand, with the validation cohort, our study highlights the significant positive correlation between hsa-let-7b-3p, hsa-miR-6826-5p, and hsa-miR-7846-3p with age. In contrast, hsa-miR-451a was negatively correlated with age. While hsa-let-7b-3p and hsa-miR-451a deregulation have been extensively reported in several tissues, hsa-miR-6826-5p and hsa-miR-7846-3p have been studied in only a few works.

hsa-let-7b-3p expression has been found in the brain [33,34] and specifically as a key regulator of multiple pathways in hippocampal samples of mesial temporal lobe epilepsy [34]. In addition, a study addressing chronological changes in the hippocampus revealed that hsa-let-7b increases in adults relative to fetal samples, suggesting an increase with age of this miRNA that is consistent with our results [35]. On the other hand, hsa-let-7b-3p was reported circulating in total plasma [36] as a predictive biomarker for amnestic mild cognitive impairment. Additionally, it was also identified circulating in extracellular vesicles (EVs) from cerebrospinal fluid (CSF) and serum “http://microvesicles.org/ (accessed in 6 August 2024)” [37]. Our results further demonstrate that as hsa-let-7b-3p expression increases with age, its target Tetratricopeptide Repeat Domain 3 (TTC33)’s expression decreases. TTC3 plays a crucial role in several cellular processes, especially related to the ubiquitin-protein transferase activity, which is integral to protein quality control (PQC) and cellular homeostasis [38]. Although the implication of TTC33 in aging has not been directly explored, a decrease in its expression could disrupt these essential functions.

hsa-miR-451a has been found in brain tissues [39,40], as well as circulating free in CSF, plasma, serum, and salivary biofluids [41,42,43,44] and also in EV-enriched samples [45,46,47]. Its presence in CSF could be indicative of brain tissue damage after a severe traumatic injury [48]. Besides, its decreased CSF circulation was reported as a biomarker of both cognitive impairment and depressive symptoms in Alzheimer’s disease patients [49] and in the serum of frail compared to robust subjects [50]. Functionally, its overexpression plays a crucial role in accelerating neuronal differentiation in vitro and in vivo [51], and its presence in EVs demonstrates a prosenescence function in human dermal fibroblasts culture, showing a role in the age-related biological processes [52].

The literature on the role of hsa-miR-6826-5p and hsa-miR-7846 is limited, and none of the articles are related to brain physiopathology. Interestingly, our study reveals an age-related increase in hsa-miR-6826-5p expression, which coincides with a decrease in its target gene, Cyp26b1. This gene encodes a member of the cytochrome P450 superfamily and functions as a critical regulator of retinoic acid levels. Cyp26b1 deficiency has been associated with impaired bone homeostasis and an unbalanced inflammatory response [53,54,55]. In addition, it has been proven that Cyp26b1 knockdown causes reduced life expectancy and systemic inflammation in mice [56]. These findings raise the intriguing possibility that the observed changes in miR-6826-5p and Cyp26b1 expression may contribute to brain aging processes through their influence on retinoic acid metabolism and related physiological pathways.

To our knowledge, this is the first work analyzing the small non-coding transcriptome in human hippocampal samples of old and centenarian individuals. We acknowledge that our work has some limitations regarding the sample analyzed. There is limited clinical information on the previous health status of the donors, which could increase the interindividual variability and include potential biological variables that might influence our results. Besides, taking into account the complexity of the hippocampus, we cannot establish whether the age-related changes in miRNA occur in the whole region or are specific to some areas of the hippocampus. Some authors report that a protein-coding gene expression gradient exists along the longitudinal axis [57] of the hippocampus, and this could also be the case for small non-coding RNAs, which should be explored in future research.

We acknowledge that our work has some limitations regarding the sample analyzed. There is limited clinical information on the previous health status of the donors, which could increase the interindividual variability. Besides, taking into account the complexity of the hippocampus.

## 4. Materials and Methods

### 4.1. Human Hippocampal Brain Samples

The discovery and validation cohort consisted of human hippocampal tissues obtained from forensic autopsy and provided by the BIOMICs group, which were part of the collection C.0000217 from the Institute of Health Carlos III Biobank register “https://biobancos.isciii.es/ListadoColecciones.aspx” (Bilbao, Spain) [17]. The sncRNAs characterization study was performed in coronal sections of hippocampal samples from 16 individuals, including young (n = 5, 27–49 years old), old (n = 8, 58–88 years old), and centenarian (n = 3, 97, 99, and 100 years old). This study was approved by the Clinical Research Ethics Committee of the Donostia University Hospital (protocol AMF-EGM-2016-01) and adhered to the tenets of the Declaration of Helsinki. Families authorized the use of these post-mortem human samples for research.

The validation cohort consisted of 118 hippocampal RNA samples composed of young (20–50 years old, n = 77), old (65–89 years old, n = 37), and centenarian (92–96 years old, n = 4) individuals. Table 1 shows the principal studied sample characteristics of discovery and validation cohorts.

### 4.2. RNA Extraction and Global sncRNA Analysis

Total RNA extraction was performed by Trizol (Life Technologies, Carlsbad, CA, USA). The sncRNAs have been profiled by Affymetrix arrays. Briefly, total RNA (200 ng) was labeled using the FlashTag HSR Biotin labeling kit (Genisphere LLC, Hatfield, Pennsylvania, PA, USA) and hybridized to the GeneChip miRNA 4.0 Array (Affymetrix, Santa Clara, CA, USA), which covers 2578, 2025 and 1996 human mature miRNAs, pre-miRNAs and snoRNAs, respectively. Labeled RNA was hybridized to the array, washed, and stained in a GeneChip Fluidics Station 450. Then, they were scanned using a GeneChip Scanner 7G (Affymetrix, Santa Clara, CA, USA). Microarray data analysis was carried out using Transcriptome Analysis Console v4.0 (TAC) software (Affymetrix, Santa Clara, CA, USA). This analysis applied the robust multi-array average (RMA) and the detection above background (DABG) algorithm, specifically tailored for human probe sets, to background correction, quantile normalization, detection, and summarization. The batch effect observed was removed using the batch effect module of the TAC software.

Finally, a differential expression analysis between the groups was performed using TAC Software to obtain the DE sncRNAs. Briefly, the software implements the limma differential expression analysis (Bioconductor package 3.19). A *p*-value < 0.05 and an absolute fold-change (FC) value ≥ |2| were considered as DE sncRNAs. The following established comparisons were considered:(1)Old vs. young;(2)Centenarian vs. old;(3)Centenarian vs. young.

Venn diagrams were built using the Venny web tool “https://bioinfogp.cnb.csic.es/tools/venny/index.html (accessed in 25 July 2024)”.

As another approach, a Pearson correlation analysis between the expression of sncRNAs and age was conducted. A Pearson correlation coefficient of r > |0.6| and a *p*-value < 0.01 were the filtering criteria.

### 4.3. Bioinformatic In Silico Analysis

The in silico analysis was conducted using the mRNA targets of the age-regulated miRNAs registered in the miRTarBase database [58]. Age-regulated miRNAs were considered those identified by differential expression and correlation analysis, and only with a miRbase identifier of a mature miRNA (excluding identifiers related to precursor miRNA and others). In addition, we used transcriptomic mRNA data available from the same hippocampal samples previously reported [17] to obtain mRNA targets that were differentially expressed in these samples, which we refer to as experimentally validated targets. With this approach, we created experimentally validated mRNA-miRNA networks in our samples using Cytoscape v3.10.1. Then, mRNA-miRNA interactions exhibiting opposite regulation were finally plotted. miRNA-mRNA opposite regulation means that when the miRNA is upregulated, the mRNA target is downregulated and vice versa. Finally, to describe the function of the mentioned mRNA targeted by the miRNAs, clusterProfiler (v4.12.2) of Bioconductor r package (v3.19) was used for the gene ontology analysis, along with human annotation database org.Hs.eg.db (v3.19.1, accessed on 30 July 2024), available through Bioconductor. Benjamini–Hochberg multiple testing FDR correction was carried out, considering the FDR value < 0.05 as a significant threshold.

### 4.4. RT-qPCR Candidates Validation

For the validation of miRNA candidates, total RNA was reverse transcribed (RT) into cDNA using a miScript II RT kit (Qiagen, Hilden, Germany). The concentration and purity of RNA were determined at 260/280 nm using a NanoDrop-ND 1000 spectrophotometer (Thermo Fisher Scientific, Wilmington, DE, USA). To quantify miRNA expression by quantitative polymerase chain reaction (qPCR), we used miScript Primer Assays and miScript SYBR Green PCR kit (Qiagen, Germany), using 10 ng cDNA following the manufacturer’s protocol. The median of hsa-let-7b-5p and hsa-miR-191-5p was used as reference genes for normalization purposes.

### 4.5. Statistical Analysis

Results of the validation cohort were expressed as ΔCq values considering the difference of Ct values between the miRNA candidate and the median of reference genes. Statistical significance was determined using a parametric ordinary one-way ANOVA test and Tukey’s post hoc test. Pearson’s correlation coefficient test was used to determine the correlation between continuous variables. Both were performed by R software v4.4.1 “http://cran.r-project.org”, and differences with *p* < 0.05 were considered statistically significant.

## 5. Conclusions

In conclusion, our findings identified a set of hippocampal miRNAs significantly deregulated with aging, highlighting their crucial role as transcription regulators. Moreover, the potential of miRNAs as non-invasive biomarkers for aging, longevity, and age-related conditions, such as frailty and neurodegenerative diseases, warrants further exploration, particularly regarding their ability to reflect tissue expression patterns when circulating freely or encapsulated in EVs.

## Figures and Tables

**Figure 1 ijms-25-12872-f001:**
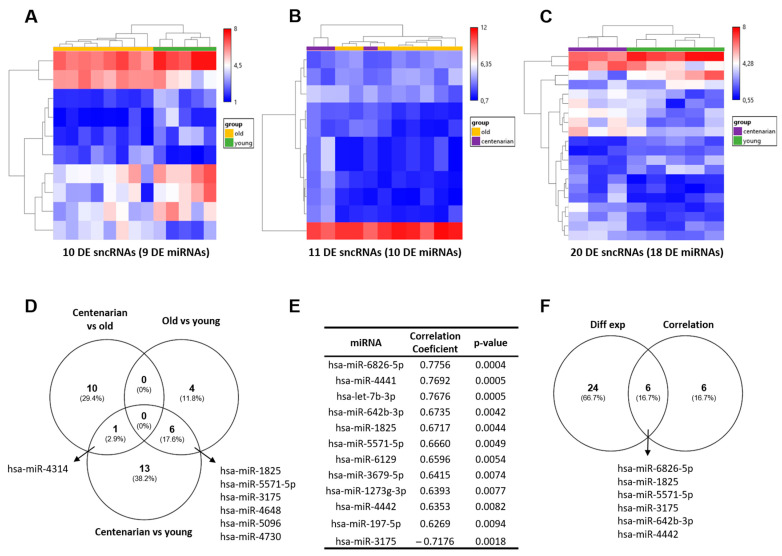
Statistically significant sncRNAs from post-mortem hippocampal RNA samples. Unsupervised hierarchical clustering of the log2 expression value of the DE sncRNA obtained between (**A**) old vs. young, (**B**) centenarian vs. old, and (**C**) centenarian vs. young. Of note, using a *p*-value < 0.05 and FC ≥ |2|, DE sncRNAs revealed are principally miRNAs. (**D**) Venn diagram showing DE sncRNAs that overlap among the comparisons. (**E**) Twelve mature miRNAs that present a significant positive (11 miRNAs) and negative (1 miRNA) correlation with age using a *p*-value < 0.01 and r > |0.6| as the cut-off. (**F**) Venn diagram of the mature miRNAs revealed by DE and correlation analyses showing overlapping. DE: differentially expressed.

**Figure 2 ijms-25-12872-f002:**
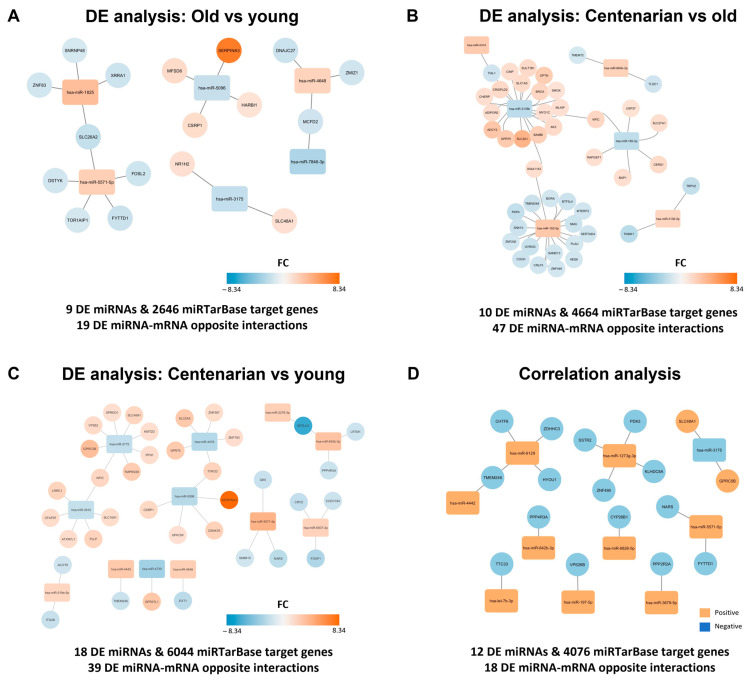
miRNA–mRNA interaction networks. miRNA and target gene networks comparing (**A**) old vs. young, (**B**) centenarian vs. old, and (**C**) centenarian vs. young groups of samples, and considering the (**D**) correlation analysis. Networks were built using experimentally validated miRNA–mRNA interactions from the miRTarBase database and published transcriptomic data from the same post-mortem hippocampal tissue samples. miRNAs and target genes are color-coded based on FC (**A**–**C**) or correlation with age (**D**).

**Figure 3 ijms-25-12872-f003:**
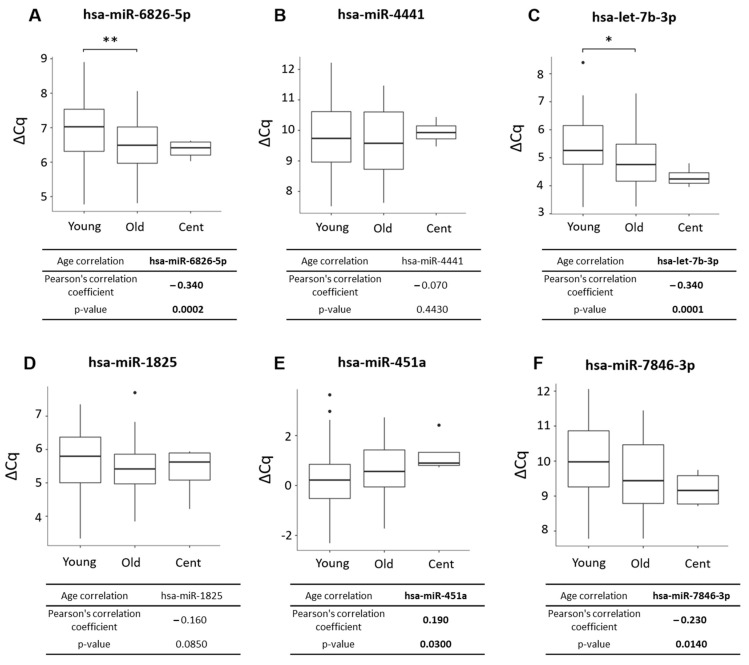
RT-qPCR validation of miRNA candidates. The miRNA candidates were validated by RT-qPCR in a second cohort of hippocampal RNA samples (n = 118). A comparative analysis between age groups and a Pearson correlation analysis with age were conducted. A significant increase in the (**A**) hsa-miR-6826-5p and (**C**) hsa-let-7b-3p expression in old when compared to young samples was observed (** *p* < 0.01 and * *p* < 0.05, respectively; ordinary one-way ANOVA test, Tukey’s post hot test). Both hsa-miR-6826-5p and hsa-let-7b-3 expression showed a statistically significant positive correlation with age. (**E**) hsa-miR-451a and (**F**) hsa-miR-7846-3p did not show significant differences between groups. In contrast, a significant negative correlation between age and hsa-miR-451a expression and a positive correlation between age and hsa-miR-7846-3p expression were found. Neither (**B**) hsa-miR-4441 nor (**D**) hsa-miR-1825 expression levels showed significant differences between the groups, and no correlation with age was found. Data are expressed as the median ± range of ΔCq from young (n = 77), old (n = 37), and centenarians (n = 4). ΔCq is the difference in Ct values between the miRNA candidate and the median of reference genes. To note, ΔCq values are inversely correlated with miRNA expression levels. A higher ΔCq means lower expression, and vice versa. ΔCq: delta quantification cycle. Those miRNAs whose correlation whith age was statistically significant were marked in bold.

**Table 1 ijms-25-12872-t001:** Information about the post-mortem brain donors.

**Discovery Cohort**		
**Age Range (Years)**	**Sex (Female/Male)**	**n**
27–49	1/4	5
58–88	1/7	8
97–100	2/1	3
**Validation Cohort**		
**Age Range (Years)**	**Sex (Female/Male)**	**n**
20–50	12/65	77
65–82	9/28	37
92–96	3/1	4

## Data Availability

The obtained data discussed in this work have been deposited in NCBI’s Gene Expression Omnibus and are accessible in GSE275442.

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
