# Peer review of "Age-Related sncRNAs in Human Hippocampal Tissue Samples: Focusing on Deregulated miRNAs"

_ijms, 2024, doi:10.3390/ijms252312872_

Round 1
Reviewer 1 Report
Comments and Suggestions for Authors
Comments to author:
The manuscript of an article, which was written by Dr. Ainhoa Alberro et al, is interesting, analyzing miRNAs from adult human hippocampus to find aging dependent expression of specific ones. However, I would suggest author to edit text and Figures. The most important is to edit Figures 3 and 4 because they are the same in concept.
Recommendation: Major revision
General comments
It seems that Figures 3 and 4 were made from the same data. Both figures merely indicate that some specific miRNAs are expressed aging dependently. If so, Figure 3 can be converted into a Table or just enough with values with essential statistical information to be added on Figure 4. Moreover, I recommend authors to indicate some data of unchanged miRNAs through aging process in the Figure 4 (delta-Cq plot). If there are such examples, it also suggests that some miRNAs are maintained at equal level in hippocampus throughout life. Alternatively, such miRNAs will be useful as standards which expression is not changed according to aging.
Specific comments
1. It would be better to add the data from samples from 50- to 70-year-old people. If there were some specific reasons to eliminate such data, it should be noted in the text.
2. In the discussion section, authors may describe possible reasons why specific miRNAs are to be up-/down-regulated in accordance with aging. Firstly, is the expression of miRNAs dependent the synthesis or degradation? Do their promoter regions have some specific motifs? Or is there any correspondence with the location on the chromosomes? If up-regulated miRNAs are transcribed from the same or similar regions, including centromeres or telomeres, of chromosomes, it should be noted.
3. If the miRNAs, including miR-6826-5p, have reference ID, it should be given in the text or a legend to Figure 4.
4. Readers will wonder how brain samples were obtained. How was biopsy executed? Or were all samples obtained from the dead? Authors know some relationships between CNS diseases and transcription of genes (page 7, L189-191). I am afraid if some samples were from CNS diseases or ADs. The information regarding each sample might better be summarized in a (supplementary) Table.
5. Additionally, hippocampus is not the homogeneous tissue as serum. If all samples were prepared from whole hippocampus, it can be compared. If some samples were excised from outer and some were from inner region, the results will be different. Authors had better comment about that.
Minor comments
P2, L48: MiRNAs; miRNAs
P2, L49: messenger RNAs target; it should be noted that intron sequences and 3’-UTRs can be targeted.
P7, L182: reveal; revealed
P8, L222: in vitro and in vivo; should be typed in italic.
Reviewer 2 Report
Comments and Suggestions for Authors
My comments to the authors are as follows:
1. The authors need to provide a demographic table providing more in-depth information about the subjects.
2. At least two studies, now a decade old, have attempted to assess miRNA expression throughout the lifespan (PMID: 23613727; PMID: 23917947). It would be interesting if the authors could discuss their findings in relation to these 2 reports.
3. A major concern about this report and the validity of the authors' results is the lack of any correction for potential demographic (biological) covariates that can confound the observed miRNA expression changes with age-related changes.
4. The study would have been stronger if they had used the larger sample for the discovery goal to minimize the potential idiosyncrasy of the very small N they have used in the microarray data.
5. The authors talk about mRNA DEG, but I failed to find any information of how regular GE was conducted.
6. It is also unclear what criteria were used to validate the computational prediction, and they also state that these have been experimentally validated, but they do not state how.
Reviewer 3 Report
Comments and Suggestions for Authors
I consider the manuscript to be well-constructed, and the data align appropriately with the applied methodology. However, the authors should include in the discussion the potential targets of the validated microRNAs and their potential impact in age process. This addition without doubt will enhance the manuscript's relevance and impact.
Round 2
Reviewer 1 Report
Comments and Suggestions for Authors
Comments to authors:
The manuscript, which was written by Dr. Ainhoa Alberro et al has been much improved from the original one. They have successfully addressed all comments that I wrote.
Recommendation: Accept as the present form
Reviewer 2 Report
Comments and Suggestions for Authors
I appreciate the authors' attempt to answer my comments. However, their responses are not encouraging to ensure that their results are consistent with their claim the reported miRNAs expressions are associated with aging. The most concerning factor is the author's own admission that these samples are forensic, with no additional information about the subjects beyond sex and age, which leaves a lot of room for potential interpretation as to why some of the observed miRNAs appear to be differentially expressed between different age groups. In the abstract, the authors even claim "...miRNAs that are deregulated with healthy aging..." without proving evidence this is true. To this end, this reviewer does not believe the result reflects the claim that any observed differential expression is a result of an age-dependent effect and not any hidden postmortem (or neuropathological) changes in the subjects.
While whether this study can be published (or not) is the editor's prerogative, my suggestion to the authors, if they want to see this published, is to be explicitly transparent about the limitations of their study and clearly state in the abstract and throughout the manuscript that their results, while they may appear interesting, lack any evidence to support the claim that these miRNAs are differentially expressed due to age-dependent effects.
Round 3
Reviewer 2 Report
Comments and Suggestions for Authors
I have no more comments to the authors